# Reporting of Incidence and Outcome of Bone Metastases in Clinical Trials Enrolling Patients with Epidermal Growth Factor Receptor Mutated Lung Adenocarcinoma—A Systematic Review

**DOI:** 10.3390/cancers13133144

**Published:** 2021-06-23

**Authors:** Anita Brouns, Safiye Dursun, Gerben Bootsma, Anne-Marie C. Dingemans, Lizza Hendriks

**Affiliations:** 1Department of Pulmonary Diseases, Zuyderland, 6162 BG Geleen, The Netherlands; a.brouns@zuyderland.nl (A.B.); g.bootsma@zuyderland.nl (G.B.); 2Maastricht University Medical Center+, Department of Pulmonary Diseases, GROW-School for Oncology and Developmental Biology, 6229 HX Maastricht, The Netherlands; safiye.dursun@mumc.nl (S.D.); a.dingemans@mumc.nl (A.-M.C.D.); 3Department of Respiratory Medicine, Erasmus MC Cancer Institute, University Medical Center Rotterdam, 3015 GD Rotterdam, The Netherlands

**Keywords:** non-small cell lung cancer, lung adenocarcinoma, epidermal growth factor mutation, bone metastases, skeletal-related events, epidermal growth factor receptor mutation, tyrosine kinase inhibitor

## Abstract

**Simple Summary:**

Around 30–60% of the patients with lung cancer develop bone metastases, which are associated with decreased survival, bone pain, and skeletal-related events such as need for radiation. Patients with an epidermal growth factor mutation (*EGFR*), a subgroup of the patients with lung cancer, seem to develop more bone metastases than other patients with lung cancer. Due to prolonged survival of these patients, they live longer with bone metastases and/or skeletal-related events, therefore optimal management is warranted. The aim of our systematic review is to gain more insight in reporting of bone metastases, skeletal-related events, and bone-specific outcome of treatment in clinical trials enrolling patients with *EGFR*-mutated lung cancer. We found that data on bone metastases and bone-related outcomes are largely lacking in clinical trials. There should be more focus on reporting and preventing of skeletal-related events in these patients.

**Abstract:**

Bone metastases, occurring in 30–60% of patients with non-small cell lung cancer (NSCLC), are associated with decreased survival, cancer-induced bone pain, and skeletal-related events (SREs). Those with an activating epidermal growth factor mutation (*EGFR*+) seem to be more prone to develop bone metastases. To gain more insight into bone metastases-related outcomes in *EGFR*+ NSCLC, we performed a systematic review on Pubmed (2006–2021). Main inclusion criteria: prospective, phase II/III trials evaluating EGFR-tyrosine kinase inhibitors, ≥10 *EGFR*+ patients included, data on bone metastases and/or bone-related outcomes available. Out of 663 articles, 21 (3176 *EGFR+* patients) met the eligibility criteria; 4 phase III (one double blind), 17 phase II trials (three randomized) were included. In seven trials dedicated bone imaging was performed at baseline. Mean incidence of bone metastases at diagnosis was 42%; 3–33% had progression in the bone upon progression. Except for one trial, it was not specified whether the use of bone target agents was permitted, and in none of the trials, occurrence of SREs was reported. Despite the high incidence of bone metastases in *EGFR+* adenocarcinoma, there is a lack of screening for, and reporting on bone metastases in clinical trials, as well as permitted bone-targeted agents and SREs.

## 1. Introduction

Activating epidermal growth factor mutations (*EGFR*) are found in approximately 10% of the Caucasian and 50% of the Asian patients with lung adenocarcinoma [1,2,3]. *EGFR* mutations are prognostic and are also predictive for efficacy of EGFR-tyrosine kinase inhibitors (TKI). For patients with advanced lung adenocarcinoma and an *EGFR* mutation, treated with epidermal growth factor receptor-tyrosine kinase inhibitors (EGFR-TKIs), the five-year survival rate is 40–50% [4,5]. This is more favorable than the historical (i.e., before the introduction of immune checkpoint inhibitors) 5.8% five-year survival rate of patients without oncogenic-driven non-small-cell lung carcinoma (NSCLC) [6]. Immune checkpoint inhibitors (ICI) as monotherapy result in disappointing outcomes in patients with EGFR-mutated lung adenocarcinoma, with low responses rates and low survival and should only be considered after exhaustion of other systemic therapies [7]. ICI combined with EGFR-TKI has been evaluated in clinical studies but the combinations were either too toxic or did not provide an advantage over EGFR-TKI alone [7].

Importantly, the biological predisposition for distant metastases seems to vary between the different molecular subgroups of non-squamous NSCLC [8,9]. The largest series is a nationwide Dutch database analysis (*n* = 2052), including all patients with metastasized non-squamous NSCLC (ns-NSCLC) at initial diagnosis with data from molecular analysis and metastasis pattern at diagnosis of stage IV disease. A significantly higher bone metastases incidence was reported in patients with an *EGFR* mutation compared with other molecular subgroups (54% vs. 33% Kirsten rat sarcoma [*KRAS*+] vs. 30.5% anaplastic lymphoma kinase fusion [*ALK*+] vs. 31.5% triple negative patients, *p* < 0.001) [9]. However, other studies (*n* = 189–1063) evaluating the incidence of bone metastases in different molecular subgroups, i.e., *EGFR*-mutated, *KRAS*-mutated, ALK-rearranged or wildtype patients with non-squamous NSCLC, showed conflicting results [8,10,11,12,13].

The currently available clinical trials evaluating bone-targeted agents (BTAs) and clinical guidelines (European Society for Medical Oncology (ESMO), National Comprehensive Cancer Network (NCCN)) providing recommendations for the management of bone metastases and skeletal-related events (SREs), do not focus on specific molecular subgroups of lung adenocarcinoma [14,15,16,17,18]. The guidelines state that it is advised to treat patients with bone metastases and a favorable survival (specified as at least three months) with BTAs. However, a more personalized advise is important, as in clinical trials especially the patients with a prolonged overall survival (OS) (i.e., historically mainly patients with metastatic breast and prostate cancer) benefit most from BTAs (i.e., significant reduction of SREs) such as bisphosphonates or denosumab [19,20,21]. As the clinical behavior of *EGFR*-mutated lung adenocarcinoma resembles metastatic breast and prostate cancer, with a real possibility for prolonged survival, data for this subgroup is also needed. However, to the best of our knowledge, clinical trials evaluating BTAs specifically in *EGFR*-mutated lung adenocarcinoma do not exist.

The risk of a negative influence on quality of life (QoL) and OS, caused by SREs, could be significant in patients with *EGFR*-mutated lung adenocarcinoma and bone metastases [22,23]. In one retrospective case-control study (*n* = 189, no use of BTAs) survival post-bone metastases diagnosis was superior for patients with an *EGFR* mutation compared to patients with a *KRAS* mutation or those without an *EGFR/KRAS* mutation, while time to first SRE was not significantly different [12]. As a result, these patients live longer with SREs. Therefore, optimal management, treatment, and outcome of bone metastases in this specific patient population is necessary and should be further evaluated.

As large prospective series on bone metastases-related outcomes are lacking for patients with *EGFR*-mutated lung adenocarcinoma, we performed a systematic review to gain more insight in the reporting of bone metastases and/or SREs, and bone-specific outcomes in patients with *EGFR*-mutated lung adenocarcinoma included in phase II/III EGFR-TKI trials. Improved knowledge about bone-related events in *EGFR*-mutated tumors can lead to better advice about the use of BTA in this subgroup of patients.

## 2. Materials and Methods

### 2.1. Search Strategy and Selection Criteria

A systematic search was performed using the Pubmed database. The search period was limited to January 2006 until January 2021 (search data 8 January 2021). The start date of January 2006 was chosen, as in 2006 EGFR-TKIs were approved by the United States of America’s Food and Drug Administration (FDA), and became standard treatment for patients with lung adenocarcinoma and an *EGFR* mutation. Published studies were identified using a search strategy based on the patient intervention comparator outcome (PICO) method (shown in Appendix A) [24]. Because the outcome variable of interest (i.e., bone metastases and SRE incidence) was an undefined endpoint in EGFR-TKI trials, we decided to exclude this outcome variable in the search strategy to prevent missing data. The Preferred Reporting Items for Systematic Reviews and Meta-Analyses (PRISMA) 2009 checklist for systematic reviews is shown in Appendix A. Furthermore, the control intervention was not included in the search strategy to include single arm EGFR-TKI trials. Trials had to include a minimum of ten patients with non-squamous NSCLC and an *EGFR* mutation, as trials in the beginning of the TKI era also included patients with wildtype *EGFR*. Only prospective, phase II and III trials were included. All inclusion criteria are summarized in Table 1.

To minimize missing articles, in the same time period, we searched for relevant articles in the meeting libraries of the American Society of Clinical Oncology (ASCO), European Society for Medical Oncology (ESMO) and International Association for the Study of Lung Cancer (IASLC).

### 2.2. Study Selection

The titles of the retrieved studies, and the abstracts of the eligible studies based on title screening, were evaluated independently by two reviewers (A.B. and S.D.). The same reviewers independently examined the full text of the remaining articles regarding the inclusion criteria. Studies were included if they met the pre-specified inclusion criteria as shown in Table 1. To complete the search, the references of all eligible articles were manually searched for additional relevant articles. In case of disagreement during study inclusion, consensus was sought.

### 2.3. Data Selection

Two reviewers (A.B. and S.D.) independently extracted relevant characteristics of each eligible study. When available and if applicable, the following data were extracted: year of publication; phase II or III trial, number of study arms, randomization method, blinding method, duration of study and follow-up, histological diagnosis, method of staging, number of patients and number of patients with an *EGFR+* mutation, intervention (i.e., type, dose, duration, route and frequency of administration of TKIs), bone metastases (i.e., incidence, outcome, treatment), SREs (i.e., incidence, outcome, treatment), secondary and primary objectives of the trial, and OS. The Jadad scale was used to assess the methodological quality of the included trials [25]. We did not perform a formal test of heterogeneity because of the heterogeneous type of trials included in the systemic review, with three quarter of the included trials being single arm (i.e., per definition high risk of bias).

## 3. Results

### 3.1. Study Selection

The literature search identified 663 unique articles in total. About 317 articles were excluded because of non-relevant titles. About 160 of the 346 remaining articles were excluded because they did not fulfill the inclusion criteria based on the abstract. The full text of the remaining 186 articles was screened; 166 articles were excluded due to: no information about bone metastases or SREs (*n* = 139), unknown *EGFR* mutational status (*n* = 10), unknown if the patients with NSCLC and bone metastases were patients with an *EGFR* mutation or if they were wildtype patients (*n* = 9), insufficient patients with an *EGFR* mutation (*n* = 4), other reasons (*n* = 4). A manual search of the reference list of the included articles revealed one additional relevant article. No additional studies were identified by searching the meeting libraries of the ASCO, ESMO, and IASLC conferences in the period 2006–2021. Ultimately, 21 articles were included in this review. The flowchart for article selection is shown in Figure 1.

### 3.2. Description of Studies

Four phase III trials [26,27,28,29], of which one double blind randomized [29], and 17 phase II trials [30,31,32,33,34,35,36,37,38,39,40,41,42,43,44,45,46], of which three randomized [37,39,40], were included. The main characteristics of the included studies are shown in Table 2.

The number of patients included in the studies ranged from 10 [38] to 556 (Flaura [29]), leading to in total 3176 patients with advanced NSCLC and an *EGFR* mutation included in this review. One trial also enrolled patients with advanced non-squamous NSCLC and *EGFR* wildtype, the results were specified per molecular subgroup [46]. The inclusion criteria were generally similar across the included trials (i.e., pathological proven locally advanced NSCLC not suitable for treatment with radical radio-chemotherapy or metastatic NSCLC). All studies, except one in which also patients with non-squamous NSCLC and *EGFR* wildtype were included [46], enrolled exclusively patients with an activating *EGFR* mutation. The other exception in inclusion criteria was systemic treatment history of the patients. The exclusion criteria concerning comorbidities were comparable among all studies. The Aura 3 was the only trial in which explicitly a statement about bone-targeted agents (BTAs) was added, it was permitted for patients to use medication (e.g., denosumab) for painful bone metastases [26]. The other trials provided no information about BTAs [26,27,28,29,30,31,32,33,34,36,37,38,39,40,41,42,43,44,45,46].

In seven trials the primary endpoint was objective response rate (ORR) on EGFR-TKI treatment [30,31,32,33,34,36] or chemotherapy combined with EGFR-TKI treatment [43]. ORR was evaluated in different patient categories: patients with NSCLC and *EGFR* mutation pretreated with chemotherapy or TKI (Aura 2 [32], [30], KCSG-Lu15-09 [31]), irrespective of previous chemotherapy [36], patients that were chemotherapy or TKI-naïve [33,34] and treatment naïve [43]. Progression-free survival (PFS) was the primary outcome in 11 trials [26,27,28,29,35,37,38,39,41,42,45]. PFS was evaluated in different patient categories: patients with NSCLC and an *EGFR* mutation who were treatment naïve in one trial [45] or treatment naïve for advanced disease in five trials (Lux-Lung 7 [40], Jo22903 and Jo25567 trial [39], Flaura, [27,29,38]). In the other trials patients with NSCLC and *EGFR* mutation were pretreated with EGFR-TKIs (Aura 3, [35,42,47]), were chemotherapy naïve (Aspiration study [41,46]) and chemotherapy naïve for advanced disease (Eurtac [28], Insight study [37]). The study of Yoshimura (2013) in which patients with NSCLC and an *EGFR* mutation who previously were treated with EGFR-TKIs and in the trial were treated with pemetrexed in combination with erlotinib or gefitinib had disease control rate as primary outcome. Two trials had co-primary endpoints: PFS and response to treatment [46] and PFS, time to treatment failure, OS in Lung-Lux 7 [40]. None of the trials had bone-metastases-related outcomes as primary or secondary endpoint.

### 3.3. Assessment of the Risk of Bias Within Studies

A formal test of heterogeneity was not performed due to the heterogeneous type of trials included in this systematic review: only four trials were randomized controlled trials, whereas two-third of the trials being single arm had per definition a high risk of bias. Instead, we used the Jadad scale to assess the methodological quality of the included studies [25]. The methodological quality of four of 21 studies were assessed as high (i.e., Jadad score ≥ 3) [27,28,29,40]. The other 17 studies were assessed as poor methodological quality (i.e., Jadad score ≤ 2) [26,30,31,32,33,34,35,36,37,38,39,41,42,43,44,45,46].

### 3.4. Results of Individual Studies

#### 3.4.1. Imaging and Incidence of Bone Metastases at Baseline

In 12 out of 21 studies the mandated imaging at study entry was described [28,30,31,32,33,36,39,42,43,44,45,46]. Dedicated bone imaging was performed in seven out of 21 trials, by means of a bone scintigraphy [33,36,39,43,44] or a 2-deoxy-2-[fluorine-18]fluoro-D-glucose positron emission tomography-computer tomography scan (FDG-PET-CT scan) [45,46].

The incidence of bone metastases at baseline was reported in 14 studies (total 1196 patients) [27,28,31,34,35,36,37,38,40,42,43,44,45,46]. Out of these 1196 patients, 502 (42%) had bone metastases at baseline (range 14–90%). In none of these studies the bone metastasis was a stratification factor.

#### 3.4.2. Imaging and Incidence of Bone Metastases during Follow-Up

In two out of 21 studies dedicated bone imaging during follow-up was performed: with FDG-PET-CT scan [45,46] or with a bone scintigraphy [33]. Bone scintigraphy was performed in the study of Reguart when clinically indicated [42].

In ten studies (total 2378 patients) the incidence of bone metastases as site of progressive disease (PD) was reported ([27,30,33,38,39,46], Flaura [29], Aura 2 [32], Aura 3 [26], Aspiration study [41]). Three to 26% of the patients had development or progression of bone metastases as site of PD (215 of 2378 patients). In none of the included studies data were provided whether bone progression was the only site of progression or not.

#### 3.4.3. Skeletal Related Events

In none of the included studies information about SREs was provided. In Table 3 a summary of the reported imaging, incidence of bone metastases and SREs is shown.

## 4. Discussion

Bone metastases with their risk of SRE development and resulting impact on QoL, can become a clinically relevant problem in patients with lung adenocarcinoma and an *EGFR* mutation because of their prolonged post-bone metastases diagnosis survival [4,12,23,48,49]. To gain more insight into bone metastases and their outcomes, we performed a systematic review focusing on screening, treatment, and reporting of bone metastases and/or SREs and bone-specific outcomes in EGFR-TKI trials.

In none of the trials, primary or secondary outcomes related to bone metastases and/or its complications were mentioned. A 42% median baseline incidence of bone metastases in patients with NSCLC and an *EGFR* mutation was reported, which is slightly lower compared to the 54% baseline incidence reported in the Dutch nationwide database study and other retrospective studies [9,12,50]. Of note, in only seven of the included trials specific bone imaging was performed at baseline, possibly resulting in an underestimation of the real incidence of bone metastases. Up to 26% of patients had progression in the bone upon PD. This probably is also underestimated as in only two of the trials standardized follow-up bone imaging was performed. Patients in the Aura 2 trial were permitted to use BTAs in case of painful bone metastases, but further information on actual BTA use and outcome was not provided. In all other trials, all data regarding BTA was lacking.

EGFR-TKIs have a high efficacy in patients with lung adenocarcinoma and an *EGFR* mutation and bone metastases [51,52]. Their efficacy in bone is mediated by blockade of receptor activator of NF-κB ligand (RANKL)-mediated osteoclast activation and by inhibiting epidermal growth factor (EGF) signaling in bone stromal cells [53]. Therefore, it could be that in this specific patient population, bone metastases do not frequently lead to SREs. Indeed, a retrospective study in patients with lung adenocarcinoma and bone metastases (*n* = 410) reported a preventive effect of EGFR-TKIs on the development of SREs: 23.5% of the patients with lung adenocarcinoma who were treated with EGFR-TKIs experienced SREs compared with 61.7% of patients without EGFR-TKI treatment (information about specific treatment in this group is not provided) (*p* < 0.001) [54]. However, even with EGFR-TKI use almost a quarter of the patients experienced SREs in this study, and in other studies, the reported frequency of SREs is even higher (37.3% to 58%) in patients with *EGFR*-mutated lung adenocarcinoma mainly treated with EGFR-TKIs [12,49,54,55,56,57]. A recently published retrospective study, which evaluated the type and frequency of SREs in patients with *EGFR*-mutated lung adenocarcinoma and bone metastases (*n* = 274, of which 148 treated with EGFR-TKI), showed that one-third of these patients developed their first SRE before start of EGFR-TKI treatment, the other two-third of the patients developed SREs in the first year of EGFR-TKI treatment [49]. The above summarized SRE percentages were observed in patients with *EGFR*-mutated lung adenocarcinoma, treated with first or second generation EGFR-TKIs. To the best of our knowledge, no data are available for the different generation EGFR-TKIs (i.e., first/second versus third) regarding efficacy specifically on bone metastases. Mouse models were set up to investigate the efficacy of osimertinib with or without bevacizumab on bone metastases of NSCLC. Treatment with osimertinib (with and without bevacizumab), showed tumor regression and bone remodeling [58]. Based on these results, it is not clear whether osimertinib is superior to earlier generation TKI in humans in the treatment of bone metastases.

In the abovementioned retrospective studies, use of BTAs varied from 0 to 65% [12,49,54,55,56,57]. Interestingly, in vitro and in vivo studies showed that bisphosphonates can act synergistically with EGFR-TKIs [54,55,59]. The in vitro study of Chang on the HCC827 NSCLC cell line expressing mutated *EGFR*, suggested that the combination of gefitinib and zoledronic acid caused more tumor suppression [59]. A small retrospective study of Cui et al. (*n* = 38) studied the efficacy of bisphosphonates in patients with *EGFR*-mutated lung adenocarcinoma and bone metastases, treated with EGFR-TKIs. They showed a significant additive effect of bisphosphonates on OS post-bone metastases diagnosis: post-bone metastases OS in EGFR-TKI + bisphosphonate group: 28.3 months versus 22.0 months in the EGFR-TKI only group, *p* = 0.0587 [55]. Another small retrospective study studied the effects of bisphosphonates in patients with *EGFR*-mutated lung adenocarcinoma and bone metastases (*n* = 62) and found comparable results (PFS and OS prolonged in the bisphosphonate + EGFR-TKI group compared with the EGFR-TKI group) [54]. As these are retrospective, small series, these data are only hypothesis generating. To the best of our knowledge, no data on denosumab combined with EGFR-TKI are available and it would be interesting to prospectively evaluate this.

Due to the increasing number of treatment options (e.g., EGFR-TKI in combination with chemotherapy, or combination of EGFR-TKI with angiogenesis inhibition), survival is further improving for patients with an *EGFR* mutation [47,51,52,60,61]. In the Flaura trial, the median OS with first line osimertinib was 38.6 months, in the NEJ009 trial (combination gefitinib with carboplatinum/pemetrexed versus gefitinib alone), median OS was 50.9 months [51,60]. Five-year survival rates for these trials have not been reported yet, but with a median OS of 50 months, 5-year survival rates resemble that of advanced breast or prostate cancer in which 28.1–30.2% of the patients are alive five years after the diagnosis [48,62,63]. This is important, as it is suggested in retrospective series that patients with *EGFR*-mutated NSCLC have a long post-bone metastatic survival of 15.5 to 28.0 months [12,49], implying that these patients live long with SREs.

Despite the similarities in the incidence and nature of *EGFR*-mutated lung adenocarcinoma and breast cancer bone metastases, current guidelines (e.g., ESMO, National Comprehensive Cancer Network [NCCN], and National Institute for Health and Care Excellence [NICE], ASCO) provide different recommendations for screening of bone metastases for different primary tumors [18,64,65,66,67]. The most important difference between the guidelines is the recommendation to screen all breast cancer patients, whereas for NSCLC only the Lung Cancer South East French Guidelines recommend to screen for bone metastases in NSCLC [18,64,65,66,67,68]. The French guideline also recommends to evaluate each bone metastasis for pain, neurological risk, and fracture risk to aid in defining the optimal bone metastasis management in harmony with the oncological treatment [68]. BTAs demonstrated benefit in reducing SREs and providing better pain control, in advanced breast patients diagnosed with bone metastases [19]. In the ESMO guideline on advanced breast cancer it is recommended to use BTAs in these patients (level of evidence I, grade of recommendation A [65]. Guidelines for lung cancer are less clear in their recommendations: the NCCN NSCLC guideline advises to consider BTAs in patients with NSCLC and bone metastases [16]. The ESMO guideline on bone health further specifies and recommends using BTAs in patients with a life expectancy of >3 months (i.e., almost all patients with an *EGFR* mutation) [18,67]. No specific recommendations for patients with *EGFR*-mutated lung adenocarcinoma were found in these guidelines. Probably because of the historically poor OS of NSCLC compared to advanced breast cancer, only 15–33% of patients with NSCLC and bone metastases are treated with BTAs in daily practice [69,70]. To the best of our knowledge no trials are ongoing that evaluate BTAs in patients with an *EGFR* mutation, although trials in patients without an oncogenic driver are ongoing [(NCT03669523 trial: denosumab in combination with nivolumab, NCT01951586 trial: denosumab in combination with chemotherapy (recently finished, results are not published)].

Drawbacks for this systematic review are: (1) The heterogeneity of the included trials with differences in populations (e.g., ethnicity) and/or follow-up which could have led to the observed differences in reported incidences of SREs; (2) the lack of primary or secondary outcomes related to bone metastases and/or related complications in studies could have led to underreporting of these outcomes.

## 5. Conclusions

Despite long (post-bone metastatic) OS of patients with *EGFR*-mutated NSCLC, and the high incidence of bone metastases in this patient population, occurrence of SRE and outcome of bone metastases is barely reported in clinical trials. Based on in vitro data found and retrospective series there might be synergistic activity of EGFR-TKI and BTA. However, prospective research is needed to validate these observations. Furthermore, the results of this systematic review stress on the importance of screening for bone metastases and reporting of clinical outcomes of treatment on bone metastases future trials for patients with *EGFR*-mutated NSCLC.

## Figures and Tables

**Figure 1 cancers-13-03144-f001:**
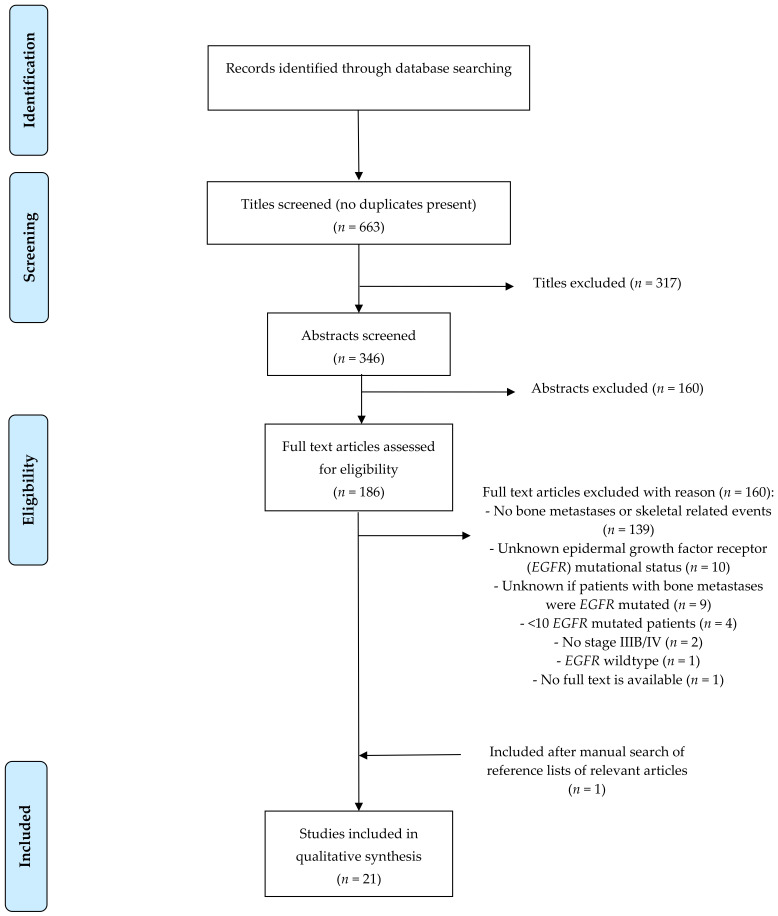
Flowchart.

**Table 1 cancers-13-03144-t001:** Inclusion criteria.

Criterion	Definition
Subjects included	Human only
Language	English
Article type	Original article; reviews excluded
Study phase	II or III
Year of publication	January 2006–Jan 2021
Site of primary tumor	NSCLC, ≥10 patients with *EGFR* mutation
Tumor stage	IIIB or IV
Age	≥18 years
Treatment	At least one of the trial arms was treatment with EGFR-TKI
Follow-up period	No lower or upper limit
Dosing, route, and frequency or duration of treatment	No restrictions
Outcome	Bone metastases and SREs at baseline or during the course of the disease, and/or their outcome. Regardless of whether they were primary, secondary, or no pre-specified endpoint of the trial

Abbreviations: NSCLC, non-small-cell lung cancer; *EGFR*, epidermal growth factor receptor; EGFR-TKI, epidermal growth factor receptor tyrosine kinase inhibitor; SREs, skeletal-related events.

**Table 2 cancers-13-03144-t002:** Main characteristics of the included studies.

Study (y)	Trial Type	Jadad Score	Total pts/*EGFR*+ pts	Histological Diagnosis (%)	Stage (%)	Treatment Arm (dose)	Comparator Arm	Median Follow-Up (Months)	Primary Objective(s)	Secondary Objective (s)
Sunaga (2007)	Phase II, single-arm, multicenter study	1	21/21	AdC (100)	IIIB (24)IV (76)	Gefitinib (250 mg q.d.)	-	12.6	ORR	PFS, tolerability
Inoue (2009)	Phase II, single-arm study	1	29/29	AdC (93) Adenosquamous (3) Undifferentiated (3)	IV (93) Other (7)	Gefitinib (250 mg q.d.)	-	17.8	ORR	PS improvement rate, toxicity, PFS, OS
Rosell (2012) [Eurtac]	Phase III, open-label, multicenter RCT	3 *	173/173	AdC (92) BAC (1) LCC (2) SCC (0.5) NOS (3)	IIIA (1) IIIB (6) IV (92)	Erlotinib (150 mg q.d.)	3-week cycles of chemotherapy ^1^	Erlotinib arm: 18.9 Chemotherapy arm: 14.4	PFS	OS, ORR, serum analysis *EGFR* mutation
Yoshimura (2013)	Phase II, single-arm, study	1	27/27	AdC (96) SCC (1)	IV (100)	3-weekly cycles of pemetrexed d1 (500 mg/m^2^) and erlotinib/gefitinib d2-16 (dose NR)	-	11.4	DCR	ORR, PFS, OS, toxicity, safety
Reguart (2014)	Phase I-II, single-arm, multicenter study	1	25/25	AdC (84) SCC (1) LCC (8)	IIIB (4) IV (96)	Erlotinib (150 mg q.d.) + vorinostat (400 mg q.d.)	-	NR	PFS at 12 weeks	Median PFS, OS
Zwitter (2014)	Phase II, single-arm, study	1	53/38	AdC (100)	IIIB (4) IV (96)	3-weekly cycles of gemcitabin 120 mg/m^2^ d1, cisplatin 75 mg/m^2^ d2, gemcitabin 1250 mg/m^2^ d4, erlotinib 150 mg q.d. d5-15	-	NR	PFS, response to treatment	OS, toxicity, metabolic response only from 2010
Yoshimura (2015)	Phase II, open-label, single-arm study	1	26/26	AdC (100)	III (4) IV (96)	3-weekly cycles of pemetrexed d1 (500 mg/m^2^) and gefitinib 250 mg q.d. d2-16	-	19.7	ORR	-
Park (2016a) [Aspiration study]	Phase II, single-arm, multicenter study	1	207/207	AdC (97) SCC (1) NOS (2)	IV (85) Recurrent (16)	Erlotinib 150 mg q.d.	-	11.3	PFS-1^2^	PFS-2 ^3^, ORR, DCR, PFS-1^2^ in exon 19 del and L858R subsets, OS, safety
Park (2016b)[Lux-lung 7]	Phase IIB, open-label, multicenter RCT	3 *	319/319	AdC (99) NOS (1)	IIIB (3) IV (97)	Afatinib (40 mg q.d.); dose escalation to 50 mg q.d. allowed after 4 weeks without AE	Gefitinib (250 mg q.d.)	27.3	PFS, time-to-treatment failure, OS	ObR, time to and duration of ObR, % pts that achieved DCR, duration of DCR, tumor shrinkage, QoL
Zwitter (2016)	Phase II, open-label, single-arm, study	1	38/38	Non-SCC (100)	IIIB (3) IV (97)	3-weekly cycles of gemcitabin (1250 mg/m^2^) d1 + 4, cisplatin 75 mg/m2 d2, erlotinib 150 mg q.d. d 5–15	-	35	PFS	-
Atagi (2016)	Combined results of 2 phase II studies: JO22903 (single-arm) and JO25567 study (randomized)	JO22903: 1 JO25567: 2	177/177	NSCLC (100)	IIIB/IV (78) Recurrent (22)	JO22903: erlotinib 150 mg q.d. JO25567: erlotinib 150 mg q.d.	JO22903: -JO25567: bevacizumab 15 mg/kg 3-weekly cycles + erlotinib 150 mg q.d.	JO22903: 20.4 JO25567: at minimum 20	PFS both studies	JO22903 and JO25567: ORR, DCR, OS. JO25567: also QoL, symptom improvement^4^, safety
Hirano (2016)	Phase II, single-arm, multicenter study	1	11/11	AdC (100)	IV (100%)	Erlotinib (25 mg q.d.); dose escalation to 150 mg q.d. in case of PD	-	NR	ORR	PFS, OS, safety
Goss (2016) [Aura 2]	Phase II, open-label, multicenter single-arm study	1	199/199	AdC (95) SCC (1) Adenosquamous (1) NOS (3)	IIIB (6%) IV (94%)	Osimertinib (80 mg q.d.)	-	13.0	ORR	PFS, duration of response, DCR, tumor shrinkage, OS, safety, QoL, pharmacokinetics
Mok (2017) [Aura 3]	Phase III, open-label, multicenter RCT	2	419/419	AdC NOS (86)	IIIB (NR) IV (NR)	Osimertinib 80 mg q.d.	3-weekly cycles of pemetrexed (500 mg/m^2^) + carboplatin (AUC 5) or cisplatin (75 mg/m^2^)	8.3	PFS	ORR, DoR, DCR, OS, tumor shrinkage, PROMS, safety, side-effect profiles
Soria (2018) [FLAURA]	Phase III, multicenter, double-blind, RCT	4 *	556/556	AdC (97) Other (3)	IIIB (5) IV (95) Missing (<1)	Osimertinib 80 mg q.d.	Erlotinib (150mg q.d.) or Gefitinib (250 mg q.d.)	15	PFS	OS, ORR, DoR, DCR, depth of response ^5^, safety
Lim (2018)	Phase II, single-arm, study	1	49/49	NSCLC NOS (100)	IV (98.7)Recurrent (10.2)	Gefitinib 250 mg q.d.	-	At minimum 6	PFS-2 ^3^	PFS-1 ^2^, difference between PFS-2- PFS-1 ^6^, OS, safety
Ahn (2019)	Combined results of 2 phase II studies (AURA extension and AURA 2 trial), both single-arm	Aura extension trial: 1 Aura 2 trial: 1	411/411	AdC (96) SCC (<1) Adenosquamous (<1) Other (3)	IIIB (4) IV (96)	Osimertinib 80 mg q.d.	-	NR	ORR	DoR, DCR, PFS, OS, safety
Zheng (2019)	Phase II, single-arm study	1	10/10	AdC (100)	IV (100%)	Erlotinib 150 mg q.d. or Gefitinib 250 mg q.d. plus thoracic radiotherapy ^7^	-	12	PFS at 12 months	PFS, OS, safety, ORR, time to progression of irradiated lesion
Cho (2019) [KCSG-Lu15-09]	Phase II, open-label, single arm, study	1	36/36	AdC (97) SCC (3)	IV (64) Recurrent (36)	Osimertinib 80 mg q.d.	-	20.6	ORR	PFS, OS, DoR, safety
Noronha (2020)	Phase III, open-label, study	3 *	350/350	Gefitinib + chemo arm: AdC (98) Adenosquamous (2) SCC (1) Gefitinib arm: AdC (97) Adenosquamous (2) SCC (1) Sarcomatoid carcinoma (1)	Gefitinib+chemo arm: IIIB (2) IV (98) Gefitinib arm: IIIB (3) IV (97)	3-weekly cycles of Gefitinib 250 mg q.d. and pemetrexed 500 mg/m^2^ + carboplatin (AUC 5) on d1, (up to four cycles), followed by 3-weekly cycles maintenance pemetrexed	Gefitinib 250 mg q.d.	17	PFS	PS, RR, toxicity, QoL
Wu (2020) [Insight study]	Phase Ib/II, open-label, study	2	55/55	Teponitinib plus gefitinib arm: AdC (97) SCC (3) Chemotherapy arm: AdC (100)	NR	Teponitinib 500 mg q.d. + gefitinib 250 mg q.d.	Pemetrexed 500 mg/m^2^ + cisplatin 75 mg/m^2^ or carboplatin (AUC 5–6) on d1 ≤ 6 cycles or 4 cycles + pemetrexed maintenance	21.8	Investigator-assessed PFS	OS, safety

Abbreviations: *EGFR*+; activating mutation in the epidermal growth factor receptor (*EGFR*), pts; patients, AdC; adenocarcinoma, q.d.; once a day, - ; not applicable, ORR; overall response rate, PFS; progression free survival, OS; overall survival, RCT; randomized controlled trial, BAC; bronchoalveolar adenocarcinoma, LCC; large cell carcinoma, SCC; squamous cell carcinoma, NOS; not otherwise specified, NSCLC; non-small-cell lung carcinoma, DCR; disease control rate, RR; response rate, PS; performance score, DoR; duration of response, ObR; objective response, QoL; quality of life, d; day, AE; adverse events, TRECIST; response evaluation criteria in solid tumors, PD; disease progression, NR; not reported, AUC; area under the curve, PROMS; patient-reported outcome measures. * High Jadad score, i.e., ≥3. ^1^ Cisplatin 75 mg/m^2^ on day 1 plus docetaxel (75 mg/m^2^ on day 1) or gemcitabin (1250 mg/m^2^ on days 1 and 8). In patients with contra-indications for cisplatin, carboplatin (AUC 6 with docetaxel 75 mg/m^2^ or AUC 5 with gemcitabin 1000 mg/m^2^) was allowed. ^2^ PFS-1; time from first study dose to first RECIST, PD or death. ^3^ PFS-2; time from first study dose to off-erlotinib PD in subset of pts who continued erlotinib therapy beyond RECIST 1.1 PD. ^4^ Measured by the Functional Assessment of Cancer Therapy-Lung (FACT-L) scale. ^5^ Defined as change in target-lesion size from baseline. ^6^ Defined as time from RECIST 1.1 progression until off-gefitinib progression. ^7^ 54–60 Gray/27–30 fractions/5.5–6 weeks.

**Table 3 cancers-13-03144-t003:** Summary of the reported imaging, incidence of bone metastases and SREs of the included studies.

Study (y)	Required Imaging at Baseline	Method of Imaging during Follow-Up	BM at Baseline (%)	BM at Progression (%)	Number of SRE (%)	BTA Use
Sunaga (2007)	Chest X-ray, chest + abdominal CT scan, brain MRI scan, radionuclide bone scan	NR	24	NR	NR	NR
Inoue (2009)	NR	NR	41	NR	NR	NR
Rosell (2012) (Eurtac)	Ct scan, optional PET-CT scan	CT scan (not further specified)	Erlotinib arm: 33 Chemotherapy arm: 33	NR	NR	NR
Yoshimura (2013)	Chest X-ray, chest + abdominal CT scan, brain MRI or CT scan, radionuclide bone scan	NR	59	NR	NR	NR
Reguart (2014)	Chest + abdominal CT scan. Brain CT scan and bone scintigraphy on indication	Chest CT scan, abdominal CT scan. Brain CT scan and bone scintigraphy on indication	40	NR	NR	NR
Zwitter (2014)	Chest X-ray, brain + chest + upper abdominal CT scan from 2010 PET-CT scan	Before 2010 NR, from 2010 PET-CT scan	63	*EGFR+* group: 26	NR	NR
Yoshimura (2015)	Chest X-ray, chest + abdominal CT scan, brain MRI or CT scan, radionuclide bone imaging or PET-CT scan	CT scan not further specified every 6 wks for first 24 wks, thereafter every 8 wks till PD or new therapy	31	NR	NR	NR
Park (2016a) (Aspiration study)	NR	NR	NR	8.2	NR	NR
Park (2016b) (Lux-lung 7)	NR	CT scan (not further specified) or MRI scan	Afatinib arm: 50 Gefitinib arm: 46	NR	NR	NR
Zwitter (2016)	PET-CT scan	PET-CT scan	63	“bone (10) most frequent site of PD.” Number of pts with PD NR.	NR	NR
Atagi (2016)	Chest + abdominal scans (CT/MRI), brain scan (CT/MRI), bone scans (bone scintigraphy, PET-CT, MRI)	NR	NR	16	NR	NR
Hirano (2016)	Chest X-ray, chest +abdominal/pelvis CT scan, brain MRI, bone scintigraphy	CT, MRI, bone scan every 2 months	NR	12.5	NR	NR
Goss (2016) (Aura 2)	CT scan or MRI scan (not further specified)	CT scan or MRI scan (not further specified)	NR	13.8	NR	Permitted, no further information
Mok (2017) (Aura 3)	Chest + abdominal scans (CT/MRI), any other areas of disease involvement based on patients’ signs or symptoms	Chest + abdominal scans (CT/MRI), any other areas of disease involvement based on patients’ signs or symptoms	NR	Osimertinib arm: 3, Platinum/ pemetrexed arm: 4	NR	NR
Soria (2018) (Flaura)	Chest + abdominal scans (CT/MRI), any other areas of disease involvement based on patients’ signs or symptoms	Chest + abdominal scans (CT/MRI), any other areas of disease involvement based on patients’ signs or symptoms	NR	Osimertinib arm: 4, Gefitinib or erlotinib arm: 4	NR	NR
Lim (2018)	NR	Tumor assessments every 8 weeks by CT-scan (not further specified)	18	NR	NR	NR
Ahn (2019)	AURA extension: CT scan or MRI scan (not further specified), AURA2 study: NR	AURA extension: CT scan or MRI scan (not further specified), AURA2 study: NR	NR	7	NR	NR
Zheng (2019)	NR	NR	90	20	NR	NR
Cho (2019) (KCSG-Lu15-09)	CT scan or MRI scan, not further specified	Chest X-ray every 3 weeks, CT scan every 6 weeks	28	NR	NR	NR
Noronha (2020)	NR	Every 9 wks by CT scans (not further specified)	Gefitinib + chemo arm: 14 Gefitinib arm: 14	Gefitinib + chemo arm: 3 Gefitinib arm: 5	NR	NR
Wu (2020) (Insight study)	NR	NR	Teponitinib plus gefitinib arm: 23 Chemotherapy arm: 37.5	NR	NR	NR

Abbreviations: BM; bone metastasis, CT scan; computer tomography scan, MRI scan; magnetic resonance scan, PET-CT scan; 2-deoxy-2-[fluorine-18] fluoro-D-glucose positron emission tomography-computer tomography scan, wks; weeks, SRE; skeletal-related event, NR; not reported.

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
