# Peer review of "Reporting of Incidence and Outcome of Bone Metastases in Clinical Trials Enrolling Patients with Epidermal Growth Factor Receptor Mutated Lung Adenocarcinoma—A Systematic Review"

_cancers, 2021, doi:10.3390/cancers13133144_

Round 1
Reviewer 1 Report
A well written and presented review article focusing on a negleted aspect of lung cancer.
Author Response
Dear reviewer,
Thank you for the critical review of our manuscript. We believe your comments made our manuscript stronger.
We obtained help with linguistic editing of the manuscript and we made minor changes throughout the text.
Reviewer 2 Report
Brouns et al. present a comprehensive report of the currently available bone metastasis data (incidence, outcome, SRE) in phase II/III trials performed in patients with EGFR lung adenocarcinoma.
The rational is clear: EGFR lung adenocarcinoma have high bone affinity and have now an expected long survival with TKI. Thus, to describe bone mets pattern and evolution and to define a bone strategy is now expected.
This systemic review is well conducted and follows the PRISMA checklist.
This review leads to the conclusion that none of these trials have correctly reported bone mets data. In that sense, even if it is a negative result, it deserves to be published.
Specific comments.
- Suggestion to replace lung cancer by lung adenocarcinoma throughout the manuscript.
- The first paragraph of the introduction is common and useless. It should be removed.
- Page 2 line 54. Add TKI avec ICI.
- In the discussion, add a limitation paragraph to gather all the difficulties encountered in this analysis (e.g : heterogeneity).
- Table 2: add a *mark to highlight the studies with a high Jadad score.
- Page 13 line 353. French guidelines recommend to screen NSCLC for bone mets (cartography) in order to check for fracture risk, pain, and neurological signs at every bone mets location (Joint Bone Spine 2019; 185). Add it.
- Page 12 line 304-308. The authors mentioned the discrepancy between studies. Can the authors explain the different reasons of such a discrepancy (prevalence in the recruited population and other points) ?
Reviewer 3 Report
The enclosed manuscript by Brouns et al is a review of the literature of bone metastases in patients with EGFR mutated lung cancer. The authors conclude that there is not sufficient data in the literature to assesss the effect of bone targeted agents in this lung cancer subset. Further studies are needed with bone specific endpoints. The data is important to better understand how to treat with form of lung cancer. I recommend acceptance with no revisions needed
Author Response

(The authors gave the same response as above.)
